# Tissue Adhesive, Self-Healing, Biocompatible, Hemostasis, and Antibacterial Properties of Fungal-Derived Carboxymethyl Chitosan-Polydopamine Hydrogels

**DOI:** 10.3390/pharmaceutics14051028

**Published:** 2022-05-10

**Authors:** Kummara Madhusudana Rao, Kannan Badri Narayanan, Uluvangada Thammaiah Uthappa, Pil-Hoon Park, Inho Choi, Sung Soo Han

**Affiliations:** 1School of Chemical Engineering, Yeungnam University, Gyeongsan 38541, Korea; msraochem@yu.ac.kr (K.M.R.); okbadri@gmail.com (K.B.N.); sanjuuthappa@gmail.com (U.T.U.); 2Research Institute of Cell Culture, Yeungnam University, Gyeongsan 38541, Korea; parkp@yu.ac.kr (P.-H.P.); inhochoi@ynu.ac.kr (I.C.); 3College of Pharmacy, Yeungnam University, Gyeongsan 38541, Korea; 4Department of Medical Biotechnology, Yeungnam University, Gyeongsan 38541, Korea

**Keywords:** mussel inspired, polydopamine, non-animal fungal mushroom carboxymethyl chitosan, hydrogel, tissue adhesive, wound dressing applications

## Abstract

In this work, fungal mushroom-derived carboxymethyl chitosan-polydopamine hydrogels (FCMCS-PDA) with multifunctionality (tissue adhesive, hemostasis, self-healing, and antibacterial properties) were developed for wound dressing applications. The hydrogel is obtained through dynamic Schiff base cross-linking and hydrogen bonds between FCMCS-PDA and covalently cross-linked polyacrylamide (PAM) networks. The FCMCS-PDA-PAM hydrogels have a good swelling ratio, biodegradable properties, excellent mechanical properties, and a highly interconnected porous structure with PDA microfibrils. Interestingly, the PDA microfibrils were formed along with FCMCS fibers in the hydrogel networks, which has a high impact on the biological performance of hydrogels. The maximum adhesion strength of the hydrogel to porcine skin was achieved at about 29.6 ± 2.9 kPa. The hydrogel had good self-healing and recoverable properties. The PDA-containing hydrogels show good antibacterial properties on *Escherichia coli* (*E. coli*) and *Staphylococcus aureus* (*S. aureus*) bacteria. Moreover, the adhesive hydrogels depicted good viability and attachment of skin fibroblasts and keratinocyte cells. Importantly, FCMCS and PDA combined resulted in fast blood coagulation within 60 s. Hence, the adhesive hydrogel with multifunctionality has excellent potential as a wound dressing material for infected wounds.

## 1. Introduction

Hydrogels are three-dimensional (3D) interpenetrating polymer networks (IPNs), structures that can absorb a huge quantity of water due to their hydrophilic structure, that are extensively employed in biomedical applications, such as tissue engineering, wound healing, drug delivery, gene delivery, bioimaging, and hemodialysis [1]. The hydrogel properties mainly depend on the functional groups in the 3D structure. The structure and mechanical properties can vary with various monomers and crosslinkers. However, the optimized concentrations of monomers and crosslinkers are necessary to obtain stable and nontoxic hydrogels for biomedical applications [2]. Wound dressings based on hydrogels are considered one of the most promising materials in wound care. They provide a moist environment with extensive exudate absorption, cooling for pain relief, antiadhesion with wound tissue, protection from secondary infections, and accelerated tissue regeneration [3]. Hydrogels with multifunctional biological properties, such as biocompatibility, biodegradability, and antibacterial and hemostatic properties, are desired to facilitate healing and improve patient comfort and outcomes.

Natural biopolymers play a vital role in wound dressing applications, as they provide good biodegradability, biocompatibility, antibacterial and hemostatic properties. Moreover, they can provide a better moist environment at the wound site to promote wound healing [4,5,6]. Chitosan (CS) is a natural polysaccharide that is composed of β-(1-4)-linked D-glucosamine (deacetylated unit) and N-acetyl-D-glucosamine (acetylated unit) [7]. It has been widely used in biomedical applications, such as tissue regeneration and wound healing applications, due to its good biodegradability, biocompatibility, and antibacterial properties [7]. However, CS has drawbacks for biomedical applications due to its low solubility in aqueous conditions and low mechanical properties. Thus, the chemical modification of CS could improve the solubility and physiological and biological properties [8]. Carboxymethyl chitosan (CMCS) is a modified CS that shows improved solubility under physiological conditions. Due to the amino groups in CMCS chains, it could react with the aldehyde functional groups of other polymers to form the dynamic Schiff base bond responsible for self-healing ability [9]. Recently, CMCS originating from non-animal sources of fungi cells of mushrooms (FCMCS) was found to have excellent physiological and biological properties, such as improved aqueous solubility in a wide range of pH solutions, good biocompatibility, and biodegradability [10,11,12]. Moreover, it can also show excellent antibacterial and hemolysis properties with rapid blood coagulation. Recently, FCMCS with antibacterial ZnO and silver nanoparticles exhibited excellent biological properties for wound dressing applications [10,11]. In another study, the films fabricated using FCMCS and polyvinyl alcohol (PVA) with multifunctional properties were beneficial for wound dressing applications [12]. Thus, FCMCS is more useful for wound dressing applications.

Although the traditional hydrogels are beneficial for wound healing, the multifunctional properties within hydrogels remain a challenge in wound healing because of poor cell affinity and tissue adhesion [3,13]. Therefore, during a surgical operation, the hydrogels cannot be fixed with the surrounding tissues. Recently, mussel-inspired chemistry based on polydopamine (PDA) provided the adhesion nature of hydrogels with good cell affinity and tissue adhesion. The adhesive component of PDA hydrogels is important for biomedical applications [14]. Its mussel-inspired PDA hydrogels can provide hemostatic properties, biocompatibility, and antibacterial property adhesion between the biological tissues. The PDA hydrogels are used in wound dressing, tissue engineering, biosensor, and bioelectronic applications [14,15,16,17]. Currently, mussel-inspired adhesive hydrogels have been developed from biopolymer-DA conjugates, such as sodium alginate (SA), hyaluronan (HA), and CS [18,19,20]. However, these hydrogels have poor mechanical properties. A PDA combination with tough hydrogels also developed within polyacrylamide (PAM) shows good wound healing properties and cell adhesion compared to other hydrogels [21,22,23]. In brief, an ideal hydrogel for wound dressing should mimic the multifunctional properties of the natural counterpart, especially its tissue adhesiveness and self-healing ability, to withstand the hydrogels at the surrounding skin and muscle tissue damage site. In addition, excellent cell affinity, tissue adhesiveness, hemostatic, and antibacterial properties enable fast wound healing. Therefore, the objective of the present study is to develop multifunctional hydrogels composed of naturally sourced FCMCS and PDA to prove the multifunctionality for wound dressing applications.

Inspired by the above-mentioned facts, we prepared multifunctional hydrogels composed via a mussel-inspired strategy. The hydrogels were prepared by incorporating an adhesive component of PDA-FCMCS within the PAM network. The hydrogels were studied for their physicochemical and biological performances to prove multifunctionality for wound dressing applications.

## 2. Materials and Methods

### 2.1. Materials

FCMCS (derived from Agaricus Bisporous Mushroom; MW = 200 KDa–2000 KDa with polydispersity 7.1; viscosity 20–1000 cps with deacetylation 80–98%) was gifted from the Endovision Company, Daegu, South Korea. Dopamine hydrochloride (DA), ammonium persulfate (APS), N,N′-methylene-bis(acrylamide) (BIS), and N,N,N′,N′-Tetramethylethylenediamine (TEMED) were purchased from Sigma-Aldrich. Acrylamide (AM) was purchased from Dae-Jung chemical metal Co., Ltd., Gyeonggi-Do, South Korea.

### 2.2. Synthesis of FCMCS-PDA Adhesive Hydrogels

A two-step method was employed for the synthesis of FCMCS-PDA-PAM hydrogels. In the first step, DA was polymerized in FCMCS solutions (2 wt%) at pH 8.5 (10 mM) for 10 min under air atmosphere. In the second step, the AM monomer (2.0 g), crosslinker (BIS, 0.05 g), and initiator (APS, 0.2 g) were sequentially dissolved in an FCMCS-PDA solution. Finally, 10 µL of TEMED was added to that solution to form a stable hydrogel. Then, the hydrogels were immersed in double-distilled water (DDW) to remove the unreacted materials. The resulting hydrogels were lyophilized for 3 days at −80 °C for further use. Appendix A lists the hydrogel formations by varying the DA content (0 wt%, 0.2 wt%, and 0.4 wt%) to AM. The formulations were labeled as FCMCS, 0.2DA-FCMCS, 0.4DA-FCMCS for 0, 0.2, and 0.4 wt% of DA respectively.

### 2.3. Characterization

The prepolymerized PDA in FCMCS solution was monitored by UV-visible spectra ranging from 200–500 nm (Shimadzu UV-2600). The functional groups within hydrogels were characterized by Fourier transform infrared spectra (FTIR; Perkin Elmer). The spectra were scanned from 4000–450 cm^−1^. The microstructure of freeze-dried hydrogels was characterized using scanning electron microscopy (SEM; Hitachi S-4800). Before analysis, the samples were coated with a platinum metal with a low deposition rate. The compressive properties of cylindrical shaped hydrogels were evaluated using a dynamic mechanical analysis (Q800 DMA; TA Instrument) in a wet state under compression mode (preload force of 0.01 N at 37 °C at a rate of 3 N/min to 18 N). The cyclic loading and unloading cycles were also performed for the 0.4DA-FCMCS-PAM hydrogel. The rheological behavior of hydrogels was analyzed using a rheometer (MCR301-Anton Paar) at room temperature under the parallel plate geometry mode (PP25; 0.5 mm gap between plates). The frequency sweep test ranged from 0.01–10 Hz at 1.0% strain amplitude. The dynamic rheological properties of the 0.4DA-FCMCS hydrogel were performed under different strain rates (1% and 1000%) at a constant frequency (1 Hz) with a fixed (60 s) time for each interval to observe the recovery behavior of hydrogels. The tensile test (Instron 3345 universal testing machine) was performed for hydrogels with a 5 kN load (crosshead speed 50 mm/min).

### 2.4. Swelling and Biodegradation Analysis

The swelling and biodegradability of hydrogels (FCMCS-PAM, 0.2DA-FCMCS, and 0.4DA-FCMCS) were evaluated by immersing samples in a phosphate-buffered saline (PBS) solution (pH = 7.4) at 37 °C for 1-, 3-, 7-, and 21-day incubation periods [16]. The dried hydrogels were weighed (W_d_). Afterward, the swollen hydrogels were removed from the vial, then the excess water attached to the surface of the hydrogel was removed using a wiper. Then, the swollen hydrogel was weighed (W_s_). Finally, the swollen hydrogel was washed with DDW and lyophilized. The lyophilized hydrogel was weighed (W_f_). The swelling ratio percentage (%SR) from these weight measurements was calculated as follows.
$$\%\mathrm{SR}=\frac{W_{s}-W_{d}}{W_{d}}\times 100\;\mathrm{Weight}\;\mathrm{loss}\;\left(\%\right)=\frac{W_{d}-W_{f}}{W_{d}}\times 100$$

### 2.5. Porosity

The porosity of the hydrogels (FCMCS, 0.2DA-FCMCS, and 0.4DA-FCMCS) was determined as described elsewhere [23]. Firstly, the hydrogels were kept in a hot air oven for 3 h at 40 °C to remove moisture. The hydrogel samples were then properly weighed (M_1_) using an electronic balance and soaked in 50 mL of a fixed quantity of absolute ethanol. After 48 h, the hydrogel samples were removed and wiped to remove surface-adhered excess ethanol with a wiper and weighed (W_2_). The following equation calculated the % porosity of hydrogel.
$$\mathrm{Porosity}\;\left(\%\right)=\frac{M_{2}-M_{1}}{\rho V}\times 100$$
where ρ is the density of the absolute ethanol and V is the hydrogel volume.

### 2.6. Cell Studies

Skin fibroblasts cells (CCCD-986sk) and keratinocytes (ATCC) were used as a model to evaluate the biocompatibility. The skin fibroblasts cells were cultured in Dulbecco’s Modified Eagle’s Medium (DMEM) with 10% fetal bovine serum (FBS) and 1% penicillin streptomycin antibiotic (PS) solution. The cells were grown in a humidified atmosphere (5% CO_2_, 37 °C).

The biocompatibility of skin fibroblasts (CCCD-986sk; ATCC) was evaluated. For this, the skin fibroblasts were grown in Dulbecco’s Modified Eagle’s Medium (DMEM) supplemented with 10% fetal bovine serum (FBS) and 1% penicillin streptomycin antibiotic (PS) solution. The cells were grown in a humidified environment (5 % CO_2_, 37 °C).

The cell viability of skin fibroblast cells and keratinocytes (5 × 10^4^ cells/cm^2^) on sterilized hydrogels (FCMCS, 0.2DA-FCMCS, and 0.4DAFCMCS) was evaluated by MTT assay in 24-well plates incubated for different time intervals (1, 3, and 7 days). The DMEM media was replaced every other day. After incubation for different durations, the medium was removed and washed with PBS. Next, 100 µL of MTT solution was added to the well plates and further incubated for 4 h in a humidified atmosphere. The formed dye crystals were dissolved in acidic isopropanol and further incubated for 30 min in a dark place. The resulting solution was analyzed for its optical density at 570 nm using a microplate reader.

The skin fibroblasts and keratinocytes (5 × 10^4^ cells/cm^2^) seeded on hydrogels were incubated for 3 days to check the cell adhesion and biocompatibility of the cells using a Live/Dead assay kit (Thermo Fisher Scientific Korea Co., Ltd., Seoul, South Korea). For Live/Dead analysis, 100 µL of Live/Dead solution (10 mL of PBS containing 20 µL of ethidium homodimer-1 and 2 µL of calcein-AM) was added to the cell-grown hydrogel samples and incubated for 30 min at room temperature. The fluorescence images (Live/Dead cells) were acquired using fluorescence microscopy (Nikon Eclipse Ti, Italy). For cell adhesion, 500 µL of 4% formaldehyde solution was added to the skin fibroblast cells grown on hydrogel samples and incubated for 10 min. Then, the hydrogels were repeatedly washed with DDW and finally dehydrated with ethanol–aqueous solution gradients. The morphology of skin fibroblasts was observed using SEM (SEM; Hitachi S-4800).

### 2.7. Hemostasis Performance (Blood Clotting Time and Hemolysis Assay)

For the blood clotting time analysis and hemolysis assay, citrated whole porcine blood was collected from a slaughterhouse (Lotte Food Co., Ltd. Ansan, South Korea). Recalcified sheep blood (100 µL) was added to the hydrogels (FCMCS, 0.2DA-FCMCS, and 0.4DA-FCMCS) fitted with 42-well plates. A well plate without hydrogel was used as a control. At specific time points, saline water was added to remove unclotted blood. The blood clotting time was measured by forming a stable blood clot at a particular time.

For the hemolysis assay, red blood cells (RBCs) were collected from citrated sheep blood as described elsewhere [23]. First, 500 µL of diluted RBCs (9 mL of saline and 1 mL of RBCs) were added to a tube containing hydrogels. Next, 100 µL of saline and 100 µL of Triton-X (0.1%) were used as positive and negative controls. Then, samples were incubated at 37 °C for 1 h. Finally, the samples were centrifuged at 3500 rpm for 10 min. The resulting supernatant solution was analyzed at 540 nm using a microplate reader to determine the hemolysis of the hydrogels. The % of hemolysis was calculated as follows.
$$\mathrm{Hemolysis}\;\left(\%\right)=\frac{\left(\mathrm{OD}\;\mathrm{of}\;\mathrm{hydrogel}-\mathrm{OD}\;\mathrm{of}\;\mathrm{negative}\;\mathrm{control}\right)}{\left(\mathrm{OD}\;\mathrm{of}\;\mathrm{positive}\;\mathrm{control}-\mathrm{OD}\;\mathrm{of}\;\mathrm{negative}\;\mathrm{control}\right)}\times 100$$

### 2.8. Antibacterial Activity

The antibacterial activities of the FCMCS, 0.2DA-FCMCS, and 0.4DA-FCMCS hydrogels were screened against Gram-positive *Staphylococcus aureus* (*S. aureus*) and Gram-negative bacteria *Escherichia coli* (*E. coli*) using the disc diffusion method. The 100 µL of bacterial culture inoculum was uniformly spread on the agar medium petri dishes with a sterile spreader for this test. The hydrogel samples (6 × 6 mm^2^) and antibiotic-loaded disc (control) were fixed on agar plates. Then, the bacterial plates were incubated at 37 °C for 12 h. The zone inhibition against *E. coli* growth around the disk was measured as the antibacterial activity of hydrogels.

### 2.9. Bacterial Adhesion

1 mL of *E. coli* and *S. aureus* bacterial solution was centrifuged and washed with PBS. Then, in 24-well plates, 200 mL of suspended bacteria in PBS were applied to the surface of sterilized hydrogels (FCMCS, 0.2DA-FCMCS, and 0.4DA-FCMCS) and incubated for 4 h at 37 °C. After incubation, the hydrogels were rinsed in PBS and then fixed overnight with 2.5% glutaraldehyde. The bacteria-adhered hydrogel samples were rinsed in water before being dehydrated using an ethanol–water combination (30%, 50%, 70%, 90%, and 100%). FE-SEM was used to examine the morphology of the *E. coli* and *S. aureus* adhered to the hydrogels (Hitachi S-4800).

### 2.10. Skin Adhesion Test

A tensile test (Instron 3345 universal testing machine) was used to determine the skin adhesion strength of hydrogels (FCMCS, 0.2DA-FCMCS, and 0.4DA-FCMCS) to porcine skin [23]. The bonding area of the hydrogels used in the tensile test was 20 × 20 mm^2^. The maximum load was divided by the bonded area to determine the strength of skin adhesion.

### 2.11. Statistical Analysis

The means and standard deviations (SD) of all the data (N = 3) are presented. ANOVA variance (one-way analysis) was used to assess the statistical differences, and a *p*-value < 0.05 was considered to be significant.

## 3. Results

### 3.1. Preparation of PDA-FCMCS-PAM Hydrogels

For wound dressing applications, a multifunctional PDA-FCMCS (cell/tissue adhesive, hemostasis, and antimicrobial) hydrogel was developed via mussel-inspired chemistry. The hydrogel preparation process is depicted in Figure 1. A two-step process accomplished the preparation of a hydrogel. In the first step, the DA monomer was polymerized in FCMCS solution under pH 8.5 conditions. The pre-gel complex was then incorporated into a covalently cross-linked PAM network with BIS using free-radical polymerization using APS as the initiator in the second step. During the formation of PDA-FCMCS hydrogel, first, the APS radicals could be involved with the polymerization of unreacted DA. The remaining APS radicals favor the formation of the PAM network. Thus, the formation of PDA-FCMCS requires a higher amount of APS. The hydrogels with different DA contents (0, 0.2, and 0.4 wt% to that of AM) were prepared, as summarized in Appendix A. In the PDA-FCMCS network, the PDA chains were linked with amine groups of FCMCS and PAM with PDA through Michael addition and Schiff base formation. In addition to that, PDA chains intertwined to form recoverable non-covalent bonds, including π-π stacking and hydrogen bonds within the PAM networks. Thus, the 0.4PDA-FMCMCS hydrogel network was hybrid cross-linked with both covalent and non-covalent bonds. Although the PDA chains are involved in the formation of Michael additions and Schiff bases with CMCS and PAM, the PDA-FCMCS network has a lot of catechol moieties, which could improve with adhesion to various surfaces. As in Figure 1, the 0.4DA-FCMCS hydrogels adhered to the different substrates, such as skin (a), leaf (b), plastic (c), rubber (d), steel (e), and glass (g). The hydrogel is stretchable without a loss of adhesion when it contacts with skin (fingers) (a and a-1) and leaf (b and b-1). The hydrogel also adhered to merged plastic tubes (c and c-1), rubber and plastic (d and d-1), and plastic and steel (e and e-1) and was able to retain adhesion capacity under 100 g of weight. The hydrogels are transparent (f-1) and adhered to the computer screen (f and f-1). Moreover, the hydrogel is easily molded with a finger (h). Hydrogel adhesion with substrates is responsible for mussel adhesion chemistry since the hydrogel contains abundant catechol groups, which are easily coordinated or covalent cross-linking with a variety of inorganic and organic substrates.

### 3.2. FTIR Spectra

The functional groups of FCMCS, 0.2DA-FCMCS, and 0.4DA-FCMCS hydrogels were analyzed for possible interactions within the hydrogel network (Figure 2 and Table 1). The FTIR-spectra of pure FCMCS powder showed characteristic peaks at 3409 cm^−1^ due to –OH and NH_2_ stretching vibrations. Peaks at 1587 and 1409 cm^−1^ were due to –COO^−^ asymmetric and symmetric stretching vibrations. Peaks at 1063 and 1410 cm^−1^ were due to C-O and C-OH functional groups. In the FTIR spectrum of FCMCS hydrogel, the peaks at 1649 and 1604 cm^−1^ were assigned to C=O stretching and N-H deformations. A peak at 1411 cm^−1^ represented the C-N stretching vibrations. For 0.2DA-FCMCS hydrogel, similar peaks belonged to PAM and FCMCS. In addition to this, a new peak at 1256 cm^−1^ was assigned to the C-N stretching of phenyl groups, which confirmed the presence of PDA in the hydrogel. With the increase in PDA concentration in the hydrogel (0.4DA-FCMCS), the peak intensity was improved, representing the existence of functional groups. Furthermore, the functional groups were shifted to lower frequencies to represent the formation of H-bonding interactions.

### 3.3. Rheology and Mechanical Performance of the Hydrogels

The rheological properties of the hydrogels (FCMCS, 0.2DAFCMCS, and 0.4DAFCMCS) showed a predominantly elastic behavior. The storage modulus (G′) and the loss tangent (tanδ = G″/G′) are shown in Figure 3a,b. The presence of PDA-FCMCS can affect the storage modulus (G′) due to an increase in viscosity. Thus, the G′ value was decreased for 0.2DA-FCMCS and 0.4DA-FCMCS hydrogels compared to FCMCS-PAM hydrogel. Furthermore, the superior elastic recovery capabilities of hydrogels with PDA were further represented by a low and steady loss factor of 0.5 (Figure 3b). Figure 3c shows the compression strength of hydrogels under a compression load of 18N. The compressive strength of the FCMCS hydrogel was 0.25 MPa. The inclusion of PDA into hydrogels is likely to result in a decrease in compressive strength due to an increase in hydrogel viscosity [23,24]. Owing to the presence of both dynamic Schiff base bonds and the Michael-type of covalent bonds between FCMCS and PDA, we expected the compressive properties and mechanical properties would increase for 0.2DAFCMCS-PAM and 0.4DA-FCMCS hydrogels compared to FCMCS hydrogel. However, the compressive strength of hydrogel was decreased to 0.15 MPa and 0.063 MPa for 0.2DA-CMCS and 0.4DA-FCMCS hydrogels from that of FCMCS (0.25 MPa). This can be explained by the fact that polymerization favored DA rather than AM. Thus, a decrease was observed in the network strength of PAM for PDA-incorporated hydrogels. The hydrogel (0.4DA-FCMCS) was further subjected to a cyclic compression test under 75% stress with five loading/unloading cycles (Figure 3d). The resulting stress–strain curves show that up to five cycles, the loading/unloading curves were identical. The hydrogel was secondary cross-linked by Michael additions and dynamic Schiff base bond as well as π-π stacking interactions. This phenomenon indicates that the network of the 0.4DA-FCMCS hydrogel dissociates and re-associates quickly, thus providing consistent hydrogel mechanical properties in the following five cycles. Typical tensile stress–strain curves with different hydrogels (FCMCS, 0.2PDA-FCMCS, and 0.4PDA-FCMCS) are shown Figure 3e. The hydrogels showed a high stretchability with remarkable tensile stress. Similar to the compression strength, the tensile stress of hydrogels were lower for 0.2DA-FCMCS (56.8 kPa) and 0.4DA-FCMCS (45.6 kPa) than for FCMCS hydrogel (72.3 kPa). However, the tensile strain was increased for 0.2DA-FCMCS (1986 %) and 0.4DA-FCMCS (3352%) hydrogels compared to FCMCS hydrogel (436%).

### 3.4. Self-Healing and Recovery Properties of Hydrogels

The PDA-FCMCS hydrogels show good self-healing and recovery properties. Hydrogel self-healing for freshly cut hydrogel is achieved by reassociation of catechol groups via the formation of dynamic Schiff bonds, π-π stacking, and hydrogen bonding interaction between polymer chains. The self-healing ability of 0.4DA-FCMCS hydrogels had healed at RT within 2 h. The hydrogel was cut into two halves and then brought into contact; they automatically rejoined within 2 h (Figure 4a). The cut line was not visible when the author stretched the self-healed hydrogel. Once the cut portion was brought into contact, dynamic Schiff bonds between the PDA-FCMCS and PAM functional groups reformed, leading to the recovery of 0.4DA-FCMCS hydrogel. The self-healed 0.4DA-FCMCS hydrogel showed a considerable strain after a self-healing duration of 24 h at RT (Figure 4d). The hydrogel had 80% recovered, with no significant differences between the self-healed and original hydrogel tensile strain and stress ratio curves. The rheology of 0.4DA-FCMCS hydrogels was performed to recover hydrogels when applied to 1000% strain followed by 1% strain for 200 s with a constant frequency (f = 1Hz) (Figure 4c). The results demonstrated that the storage modulus was decreased from 1000 to 600 Pa when a strain was applied from 0.1% to 1000% and then recovered its original position within min, and then constant G′ was attained up to 200 s. The hydrogel was further subjected to a dynamic rheological study to know the self-recovery property at different stains (1% and 1000%) (Figure 4b). The results showed that the hydrogel networks were stable at 1% (G′ > G″), and then, by switching the strain to a large amount of strain (1000%), the hydrogel networks were destroyed (G″ > G′). Then, switching the strain to 1%, the G′ and G″ were recovered to their original values (G′ > G″), confirming the recovery properties of the hydrogel. A stable recovery trend was followed for different cycles, representing the good recovery property of 0.4DA-FCMCS hydrogel.

### 3.5. Microstructure of Hydrogels

An SEM was used to examine the microstructure of the freeze-dried hydrogels (Figure 5). The CMCS-PAM hydrogel exhibited a porous structure with fibers that were uniformly interconnected. The formation of fibrous structures was attributed to the H-bonding interactions between –NH_2_ and –COOH groups within the FCMCS polymer chains during the formation of a hydrogel. In the presence of PDA, the PDA-FCMCS hydrogels had well-interconnected microporous three-dimensional microfibrils, indicating PDA chains were well connected with CMCS and PAM. PDA complexation may cause the microfibrils due to the formation of π-π and hydrogen bonding interactions of PDA functional groups. Interestingly, the microfibrils were formed along with FCMCS fibers due to covalent bonding between PDA and FCMCS (Appendix A). The average pore sizes of FCMCS, 0.2DA-FCMCS, and 0.4DA-FCMCS hydrogels were 72.4 + 0.68, 80.4 + 0.90, and 83.7 + 0.60, respectively (Figure 6a). The increase in DA content increases the pore size of hydrogels. The porosity of hydrogels also improved by increasing the PDA. Hydrogels with a high porosity can help nearby cells migrate into the wound and repair it. The porosity of the hydrogels was investigated for this purpose. The porosities of FCMCS, 0.2DA-FCMCS, and 0.4DA-FCMCS hydrogels were (83.5 + 1.5), (88.2 + 0.95), and (91.6 + 1.6), respectively (Figure 6b). Therefore, the hydrogels with highly interconnected pores with good porosity may help the potential of cells in the growth, adhesion, proliferation, and movement of nutrients and oxygen molecules.

### 3.6. Swelling and Biodegradation of Hydrogels

Traditional wound-healing biomaterials can operate as a temporary barrier to stop bleeding, prevent infection, and promote regeneration. Because of their ability to maintain moisture, hydrogels are among the greatest prospects for wound healing. The swelling of hydrogel capacity is critical for wound dressing applications. PDA-FCMCS hydrogels have a large amount of water and a hydrophilic nature in wound dressing. Exudate absorbing efficacy is important when choosing an appropriate dressing to provide a proper moist environment to prevent the wound bed from drying out and infection. The PDA-FCMCS hydrogel has good swelling ability and biodegradation properties (Figure 7a,b). The swelling ratios of FCMCS, 0.2DA-FCMCS, and 0.4DA-FCMCS hydrogels reached equilibrium after 3 days at 37 °C in PBS solution. The DA-containing hydrogels had good swelling ratios compared to FCMCS hydrogels. Furthermore, because of its larger pore size and catechol groups in the PDA, the 0.4DA-FCMCS hydrogel had the highest swelling ratio of all the hydrogel samples. The biodegradation property plays a significant role in wound dressing for biomedical applications. The hydrogels (FCMCS, 0.2DA-FCMCS, and 0.4DA-FCMCS) were degradable with respect to the incubation time period (Figure 7b). Following a 21-day incubation, the hydrogels showed a degradation trend of FCMCS > 0.2DAFCMCS > 0.4DAFCMCS, with weight losses of 12.8, 11.6, and 8.4%, respectively. The combination of PAM and PDA with biodegradable FCMCS within hydrogel networks maintains biodegradation. Therefore, the hydrogels provide a good environment for the absorption of moisture for wound healing [23,25].

### 3.7. Cell Study

The catechol groups in PDA show good cell proliferation, viability, and adhesion in both skin fibroblast cells for biocompatibility. The PDA-FCMCS hydrogel had cell affinity to skin fibroblast cells because catechol functional groups existed in the hydrogel, as they provide strong connections to the cell membranes (thiol and imidazole) compared to other phytochemicals, such as tannic acid (TA) and gallic acid (GA).

Fibroblast cells can play an important part in the wound healing process. The invasion of fibroblasts results in the formation of a fibrin matrix, which, when combined with HA, accelerates wound healing. As a result, HA may play a key role in fibroblast cell migration, which provides ECM for fast wound healing. Keratinocytes are the main cellular component of the epidermis and play a number of important roles in wound healing. They are engaged in the complex mechanisms of initiation, maintenance, and completion of wound healing [23]. Therefore, skin fibroblasts and keratinocytes cells were used as model cells for the cell adhesion and viability of hydrogels. The cell viability of hydrogels was analyzed using an MTT assay. The cell viability was maintained for 1, 3, and 7 days for hydrogels (Figure 8a,b). The PDA-containing hydrogel possesses a good proliferation of skin fibroblasts and keratinocytes cells. Further, the cell viability of hydrogels with skin fibroblasts and keratinocytes was analyzed using fluorescence microscopy. Figure 9 shows skin fibroblast cells on the hydrogels (0.2DA-FCMCS, 0.4DA-FCMCS, and FCMCS-PAM) to spread with expanded filopodia within 3 days of culture. Similar to skin fibroblasts, the keratinocytes cells also grew well in hydrogels. PDA-containing hydrogels had good viability without dead cells. Moreover, the attachment of cells on the hydrogel scaffold was analyzed using SEM. As in Figure 10, PDA-containing hydrogels (0.2 and 0.4DA-FCMCS) with skin fibroblasts and keratinocyte cells showed a good cell adhesion compared to FCMCS hydrogels due to the catechol groups in the PDA, which gives good adhesion of cells on the hydrogel. Overall, the PDA-containing hydrogels show good cell viability and adhesion compared to those without DA content; due to the adhesive material of DA, it gives a good environment to cells and plays an important role in wound dressing applications.

### 3.8. In Vitro Blood Clotting Study

The blood clotting time mainly explains the hemostatic ability of a hydrogel to persuade thrombosis in whole porcine blood using an evaluation of the performance of a parameter called blood clotting time. The in vitro blood clotting capabilities of FCMCS, 0.2DA-FCMCS, and 0.4DA-FCMCS hydrogels were evaluated by monitoring the whole porcine blood clotting times (Figure 11a,b). The hydrogel was inserted on a 24-well plate, and whole porcine blood was added to both the hydrogel-containing well plate and the control well plate. In general, the blood clotting time was reported as 300–600 s. Similarly, the whole porcine blood clotting time was 360 s. Originally, the FCMCS showed good blood coagulation properties. In this study, the fibrous structure of FCMCS in the hydrogel provided blood coagulation of whole porcine blood. Furthermore, PDA fibrils that formed along the FCMCS fibers improved blood coagulation. The blood clotting time was decreased to 240 s. The tissue adhesive hydrogels incorporated with PDA showed 180 s and 60 s blood clotting times for 0.2DA-FCMCS and 0.4DA-FCMCS hydrogels. The results showed that adding PDA to the hydrogel significantly increased its ability to clot blood. This is because of the formation of PDA fibrils and FCMCS fibrous structures in which the catechol groups involve the coagulation of blood [26].

### 3.9. In Vitro Hemolysis Study

The hemocompatibility of the FCMCS, 0.2DA-FCMCS, and 0.4DA-FCMCS hydrogels was analyzed using the hemolysis assay performance (Figure 11c,d). As in Figure 11, the supernatant of the hydrogel samples treated with RBCs was clear and is similar to saline-treated diluted RBCs. The supernatant solution of Triton-X-treated RBCs was unclear because of the rupture cell membranes, which led to the release of hemoglobin from RBCs. All hydrogel samples had hemolysis at less than 3%, which is well below the acceptable range of 5% for biomaterials [16,26]. Among all hydrogels, 0.4DA-FCMCS hydrogel displayed high hemocompatibility because a higher amount of PDA in the hydrogel provided strong electrostatic repulsion interactions between the PDA of the hydrogel and lipid bilayers, thereby limiting contact with RBCs [27].

### 3.10. Antibacterial Activity

Figure 12a,b illustrate the antibacterial activity of hydrogels against *E. coli* and *S. aureus* bacteria. The results indicated a small increment in zone inhibition around the hydrogel disk, with increased DA content in the hydrogel. The zone diameter for FCMCS, 0.2DA-FCMCS, and 0.4DAFCMCS hydrogels were 6.2, 6.8, and 7.6 mm for *E. coli* and 6.1, 6.5, and 7.3 mm for *S. aureus* bacteria, respectively. Compared to control, a small inhibition growth diameter formed around the disc represents the network structure of the hydrogel. In this work, the antibacterial hydrogels were based on the contact between the active surface and bacteria without releasing antibacterial agents. The improved antibacterial response of hydrogels with PDA is attributed to the free catechol groups on hydrogel surfaces.

### 3.11. Bacterial Adhesion

SEM images of bacteria-seeded hydrogels were used to evaluate bacterial adhesion. The bacterial adhesion of hydrogels to *E. coli* and *S. aureus* is shown in Figure 12c,d. Due to the antiadhesive bacterial characteristic of FCMCS, there were less bacteria adhered to the surface of the FCMCS hydrogel, resulting in antiadhesive properties towards both bacteria (*E. coli* and *S. aureus*). Increased PDA content in the hydrogel increased bacterial adhesion significantly. The catechol hydroxyl groups in the hydrogel networks, which may create contacts with the bacterial cell membranes, are responsible for the strong bacterial adhesion of the 0.4DA-FCMCS hydrogel with both bacteria (*E. coli* and *S. aureus*) [28].

### 3.12. Skin Adhesion Test

Skin adhesion is important for wound dressing applications, as they adhere to the skin during the wound dressing without any adhesive tapes and stretch. For this, the adhesion test of hydrogel was performed with porcine skin. The presence of catechol functional groups on polymer chains was predicted to allow faster penetration and greater adherence to porcine skin tissue surfaces. The quantity of catechol functional groups in the hydrogel significantly impacted the hydrogel’s ability to adhere to porcine skin. FCMCS, 0.2DA-FCMCS, and 0.4DA-FCMCS hydrogels had skin adhesion strengths of 16.9 + 1.6 kPa, 21.6 + 2.4 kPa, and 29.6 + 2.9 kPa, respectively (Figure 12e). Hence, the hydrogel skin adhesion strength to porcine skin was increased with the increasing amount of DA and reached a maximum value of 29.6 ± 2.9 kPa with a 0.4 wt% of DA/AM ratio. Free catechol groups on PDA interact with –NH_2_ or –SH groups on the skin tissue surface, which causes stronger skin adhesion. [23].

## 4. Conclusions

In summary, we created antibacterial wound dressing skin adhesive hydrogels based on PDA, FCMCS, and PAM. In PDA hydrogels, FCMCS was used to provide broad-spectrum antibacterial activity while maintaining mechanical characteristics. The hydrogel had highly interconnected porous structures along with PDA-FCMCS fibrils. The hydrogel was self-healing and self-recoverable due to the numerous connections. In vitro antimicrobial tests, cell affinity tests, and hemostasis tests all showed that the hydrogel might be employed as a wound dressing material for wound healing. Overall, with PDA and FCMCS, the hydrogel had multiple functionalities, such as tissue adhesive, biocompatible, self-healing, and antibacterial properties. Considering the multifunctional properties, the PDA-FCMCS hydrogel could enhance the wound dressing abilities for practical applications.

## Data Availability

All available data are reported in the article.

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
