# Peer review of "Tissue Adhesive, Self-Healing, Biocompatible, Hemostasis, and Antibacterial Properties of Fungal-Derived Carboxymethyl Chitosan-Polydopamine Hydrogels"

_pharmaceutics, 2022, doi:10.3390/pharmaceutics14051028_

Round 1

Reviewer 1 Report

This is a well-organized and well-illustrated paper, has an important clinical message, and should be of great interest to the readers. The article reported a new type of hydrgel for wound dressing applications and deserves publication after some minor revisions.

  1. Line 100, 'MW=200 KDa-2000 Kda' to 'Mw=200 KDa-2000 KDa'. The polydispersity (PDI) of FCMCS should be added.
  2. Line 154-155, '(M2-M1' to 'M2-M1'. 
  3. Line 197-198, 'X100' to '×100%'.
  4. Line 108-109, 'In the first step, DA was polymerized in FCMCS solutions at pH 8.5 for 10 min under air atmosphere.' The concentration of  FVMCS, and the salt concentration of the buffer solution should be added. Otherwise, it will be hard to be repeated by others.
  5. The microstructure of hydrogels checked by SEM is hard to be believed due to the drying effect.
  6. It is not clear why 0.4DA-FCMCS has the best hemocompatibility among FCMCS, 0.2DA-FCMCS, and 0.4DA-FCMCS.
  7. Reference. Line 76-78, several recent studies (doi.org/10.1021/acs.langmuir.1c00364; doi.org/10.1016/j.actbio.2018.11.015; doi.org/10.1016/j.colsurfb.2022.112409) related to this point should be included.

Author Response

Line 100, 'MW=200 KDa-2000 Kda' to 'Mw=200 KDa-2000 KDa'. The polydispersity (PDI) of FCMCS should be added.
Response: Thank you for your comment. Now we have provided the PDI in the materials section. 

Line 154-155, '(M2-M1' to 'M2-M1'. 
Response: Thank you for your comment. Now we have corrected the equation. 

Line 197-198, 'X100' to '×100%'.
Response: Thank you for your comment. Now we have corrected the equation. 

Line 108-109, 'In the first step, DA was polymerized in FCMCS solutions at pH 8.5 for 10 min under air atmosphere.' The concentration of  FVMCS, and the salt concentration of the buffer solution should be added. Otherwise, it will be hard to be repeated by others.
Response: Thank you for your comment. Now we have provided. 

The microstructure of hydrogels checked by SEM is hard to be believed due to the drying effect.
Response: Thank you for your comment. We have used freeze dry for analysis of microstructure of hydrogels. Previous literature also proved the microstructure of the hydrogels using freeze drying technique. We believe that, this freeze dry is suitable for creating porous structures for cell growth. 

It is not clear why 0.4DA-FCMCS has the best hemocompatibility among FCMCS, 0.2DA-FCMCS, and 0.4DA-FCMCS.
Response: Thank you for your comment. Now we have provided clear information. 

Reference. Line 76-78, several recent studies (doi.org/10.1021/acs.langmuir.1c00364; doi.org/10.1016/j.actbio.2018.11.015; doi.org/10.1016/j.colsurfb.2022.112409) related to this point should be included.
Response: Thank you for your comment. Now we have cited the articles.

Reviewer 2 Report

The manuscript can be published after some revisions:

Most important - the authors should downplay their claims regarding "wound dressing applications". They do not show any data on wound dressing, hence the claim is a mere speculation. The paper itself is OK and eventually they may indeed use these materials for wound dressing. However, currently this is not show, so the title and the claims in the text need to be changed.  

Figure 1 is not informative, the individual images are not seen well, please consider revising this figure

Line 347-348 "the microfibrils were formed along with FCMCS fibers due to covalently bonding between PDA and FCMCS" can the authors comment on how they determined that the covalent bonding occurs?

Please add visible scale bars to Fig 8

What was the reason for using fibroblasts cells for in vitro study? Why no other cells were studied? Same applies for choosing E. coli as the only bacteria.

Author Response

Most important - the authors should downplay their claims regarding "wound dressing applications". They do not show any data on wound dressing, hence the claim is a mere speculation. The paper itself is OK and eventually they may indeed use these materials for wound dressing. However, currently this is not show, so the title and the claims in the text need to be changed.  

Response: Thank you for your suggestion. Now we have changed the title of the manuscript.

 Figure 1 is not informative, the individual images are not seen well, please consider revising this figure

Response: Thank you for your suggestion. Now we have revised the figure.

Line 347-348 "the microfibrils were formed along with FCMCS fibers due to covalently bonding between PDA and FCMCS" can the authors comment on how they determined that the covalent bonding occurs?

Response: Thank you for your comment. In general PDA based hydrogels show PDA fibrils. In our work, we have used FCMCS. Its having amino groups. As per PDA chemistry with amine groups, amino groups can easily formed covalent bond with PDA. We expect that FCMCS can easily bond with PDA fibrils. We assume that t the microfibrils were formed along with FCMCS fibers due to covalently bonding between PDA and FCMCS.

Please add visible scale bars to Fig 8

Response: Thank you for your comment. Now we have provide the scale bars to the Fig 8.

What was the reason for using fibroblasts cells for in vitro study? Why no other cells were studied? Same applies for choosing E. coli as the only bacteria.

Response: Thank you for your comment. Now we have done two cell line such as skin fibroblasts and keratinocytes. In addition we have also provided the data for antibacterial activity of hydrogels using both Gram-positive (S.aureus) and Gram-negative bacteria (E.coli).

Reviewer 3 Report

I have gone through this manuscript and have suggestions and comments to improve the readability and quality of the manuscript.

  • Abstract needs to be more concise and focused on the finding of the utilization of FCMCS-PDA, I am surprised the author mentioned in the abstract that the newly designed hydrogels have good adhesion properties such as leaf and computer screen (how it is relevant to this paper?).
  • The introduction needs to be more focused and describe the rationale of the study. At present too many general statements, it will be great if the author will focus on the objectives and related statements.
  • The synthesis of multifunctional hydrogels is not explained well, it will be great if the authors could give more details mechanisms in the schematic diagram.
  • The characterization of the newly designed materials needs to explain in more detail.
  • It will be nice if the author could mention how the swelling study of this manuscript is relevant to wound healing applications.
  • Porosity and why only skin fibroblast relevance in wound healing needs to be described in detail
  • Hemolysis and blood clotting time experiments need to be clearer and add some recent references. It is confused between procaine and sheep blood, please clarify it
  • During the antibacterial study what type of Gram-negative bacteria Escherichia coli (coli) you have used and why you have selected Gram-negative bacteria needs to be explained.
  • The conclusion needs to be more focused and explained the finding of the study.
  • It will be nice if you could add some recent related references from prestigious pharmaceutics journal (MDPI)

Author Response

  • Abstract needs to be more concise and focused on the finding of the utilization of FCMCS-PDA, I am surprised the author mentioned in the abstract that the newly designed hydrogels have good adhesion properties such as leaf and computer screen (how it is relevant to this paper?).

Response: Thank you for your comment. Now we have revised the abstract. In the abstract we have removed the sentence of adhesion properties of various substrates.

  • The introduction needs to be more focused and describe the rationale of the study. At present too many general statements, it will be great if the author will focus on the objectives and related statements.

Response: Thank you for your comment. Now we have revised the introduction part with statements.

  • The synthesis of multifunctional hydrogels is not explained well, it will be great if the authors could give more details mechanisms in the schematic diagram.

Response: Thank you for your comment. We have already provided the schematic representation of mechanism of hydrogel formation in supporting information.

  • The characterization of the newly designed materials needs to explain in more detail.

Response: Thank you for your comment. Now we have revised.

  • It will be nice if the author could mention how the swelling study of this manuscript is relevant to wound healing applications.

Response: Thank you for your comment. Now we have mentioned the importance of swelling study in wound healing applications.  

  • Porosity and why only skin fibroblast relevance in wound healing needs to be described in detail

Response: Thank you for your comment. Now we have provided cell viability of keratinocyte cells and discussed the importance of cells in wound healing.

  • Hemolysis and blood clotting time experiments need to be clearer and add some recent references. It is confused between procaine and sheep blood, please clarify it

Response: Thank you for your comment. Now we have provided clear discussion with recent references. We have clarified the blood source.

  • During the antibacterial study what type of Gram-negative bacteria Escherichia coli (coli) you have used and why you have selected Gram-negative bacteria needs to be explained.

Response: Thank you for your comment. Now we have provided Gram-positive bacteria results and discussed.

  • The conclusion needs to be more focused and explained the finding of the study.

Response: Thank you for your suggestion. Now we have revised conclusion.

  • It will be nice if you could add some recent related references from prestigious pharmaceutics journal (MDPI)

Response: Thank you for your suggestion. Now we have provided the some recent references from MDPI pharmaceutics journal.

Reviewer 4 Report

Comments

Title: Multifunctional hydrogels from fungal mushroom-derived carboxymethyl chitosan-polydopamine hydrogels via mussel-inspired chemistry for wound dressing applications

The work presented in this article is interesting and meets sufficient data of interest to the readers of Pharmaceutics. It represents a relatively study about adhesive hydrogel as a wound dressing application.

I have some major comments for the authors below:

  1. Please edit the writing in terms of readability and grammatical problems.
  2. For FT-IR results, please prepare a table and summarize the results in one table including wavelength, functional group, and explanation.
  3. Please adjust the font size of the figures based on text.
  4. The statistics are missing. Please add appropriate statistics to the graphs. P-value and SD should be reported.
  5. Please reconsider about units. “37oC” is not the correct form.
  6. Cell proliferation slope is too narrow. 50,000 cell/cm2 is too much for cell proliferation. The cell proliferation test should be repeated based on previous literature.
  7. Scale bare is required for figures 8 b, c, and d.
  8. The results from Figures 8 a and 8 b, c, and d are mismatching.
  9. SEM from cells should be magnified for cell morphology study.
  10. LDH assay for evaluation of cytotoxicity is required.
  11. Control without hydrogel is required for cell study.
  12. Clot adhesion to the hydrogel should be evaluated.
  13. Error bars should be added to graphs as standard deviation.
  14. Antibacterial activity should be evaluated using both gram-positive and negative models. Samples should be in round shape and the same size.
  15. Bacterial adhesion to the hydrogel should be evaluated.
  16. Conclusion does not support all results.

Author Response

Please edit the writing in terms of readability and grammatical problems.

Response: Thank you for your comment. Now we have corrected grammatical errors.

For FT-IR results, please prepare a table and summarize the results in one table including wavelength, functional group, and explanation.

Response: Thank you for your comment. Now we have summarized the FTIR peak details in the Table 1.

Please adjust the font size of the figures based on text.

Response: Thank you for your comment. Now we have corrected figures with font size.

The statistics are missing. Please add appropriate statistics to the graphs. P-value and SD should be reported.

Response: Thank you for your comment. Now we have provided the P-value and SD values.

Please reconsider about units. “37oC” is not the correct form.

Response: Thank you for your comment. Now we have corrected.

Cell proliferation slope is too narrow. 50,000 cell/cm2 is too much for cell proliferation. The cell proliferation test should be repeated based on previous literature.

Response: Thank you for your comment. Based on previous literature we have performed the cell study. Most of the reports showed that, they have used 50,000 cells/cm2. For our study we have used same. 

Scale bare is required for figures 8 b, c, and d.

Response: Thank you for your comment. Now we have provided the scale bars in Fig. 8.

The results from Figures 8 a and 8 b, c, and d are mismatching.

Response: Thank you for your comment. Now we have corrected.

SEM from cells should be magnified for cell morphology study.

Response: Thank you for your comment. We are sorry for this. At this time we are not able to provide the magnified images.

LDH assay for evaluation of cytotoxicity is required.

Response: Thank you for your comment. We are sorry for this. At this time we are not able to do the experiment. We will provide our future experiments.

Control without hydrogel is required for cell study.

Response: Thank you for your comment. Now we have provided the control of the cell culture study.

Clot adhesion to the hydrogel should be evaluated.

Response: Thank you for your comment. We are sorry for this. At this time we are not able to repeat the experiment. We will provide our future experiments.

Error bars should be added to graphs as standard deviation.

Response: Thank you for your comment. Now we have provided the error bars.

Antibacterial activity should be evaluated using both gram-positive and negative models. Samples should be in round shape and the same size.

Response: Thank you for your comment. As per your suggestion, now we have provided the results of antibacterial activity of hydrogels with both Gram-positve and Gram-negative bacteria with round shaped samples.

Bacterial adhesion to the hydrogel should be evaluated.

Response: Thank you for your comment. Now we have provided the SEM images of bacterial adhesion.

Conclusion does not support all results.

Response: Thank you for your comment. Now we have revised the conclusion.

Round 2

Reviewer 2 Report

The revised manuscript can be accepted 

Reviewer 4 Report

Thanks to providing the the revision.